# Proteomics Profiling of Stool Samples from Preterm Neonates with SWATH/DIA Mass Spectrometry for Predicting Necrotizing Enterocolitis

**DOI:** 10.3390/ijms231911601

**Published:** 2022-10-01

**Authors:** David Gagné, Elmira Shajari, Marie-Pier Thibault, Jean-François Noël, François-Michel Boisvert, Corentin Babakissa, Emile Levy, Hugo Gagnon, Marie A. Brunet, David Grynspan, Emanuela Ferretti, Valérie Bertelle, Jean-François Beaulieu

**Affiliations:** 1Laboratory of Intestinal Physiopathology, Faculty of Medicine and Health Sciences, Université de Sherbrooke, Sherbrooke, QC J1H 5N4, Canada; 2Centre de Recherche du Centre Hospitalier Universitaire de Sherbrooke, Sherbrooke, QC J1H 5N4, Canada; 3Department of Immunology and Cell Biology, Faculty of Medicine and Health Sciences, Université de Sherbrooke, Sherbrooke, QC J1H 5N4, Canada; 4PhenoSwitch Bioscience Inc., 975 Rue Léon-Trépanier, Sherbrooke, QC J1G 5J6, Canada; 5Department of Pediatrics, Faculty of Medicine and Health Sciences, Université de Sherbrooke, Sherbrooke, QC J1H 5N4, Canada; 6Research Center, Centre Hospitalier Universitaire Ste-Justine, Université de Montréal, Montréal, QC H3T 1C5, Canada; 7Department of Pathology and Laboratory Medicine, Faculty of Medicine, University of British Colombia, Vancouver, BC V6T 2B5, Canada; 8Division of Neonatology, Department of Pediatrics, Children’s Hospital of Eastern Ontario (CHEO) and CHEO Research Institute, Ottawa, ON K1H 8L1, Canada; 9Division of Neonatology, Department of Pediatrics, Faculty of Medicine and Health Sciences, Université de Sherbrooke, Sherbrooke, QC J1H 5N4, Canada

**Keywords:** necrotizing enterocolitis, biomarker, diagnosis, prediction, proteomics, stool analysis, SWATH, DIA mass spectrometry

## Abstract

Necrotizing enterocolitis (NEC) is a life-threatening condition for premature infants in neonatal intensive care units. Finding indicators that can predict NEC development before symptoms appear would provide more time to apply targeted interventions. In this study, stools from 132 very-low-birth-weight (VLBW) infants were collected daily in the context of a multi-center prospective study aimed at investigating the potential of fecal biomarkers for NEC prediction using proteomics technology. Eight of the VLBW infants received a stage-3 NEC diagnosis. Stools collected from the NEC infants up to 10 days before their diagnosis were available for seven of them. Their samples were matched with those from seven pairs of non-NEC controls. The samples were processed for liquid chromatography-tandem mass spectrometry analysis using SWATH/DIA acquisition and cross-compatible proteomic software to perform label-free quantification. ROC curve and principal component analyses were used to explore discriminating information and to evaluate candidate protein markers. A series of 36 proteins showed the most efficient capacity with a signature that predicted all seven NEC infants at least a week in advance. Overall, our study demonstrates that multiplexed proteomic signature detection constitutes a promising approach for the early detection of NEC development in premature infants.

## 1. Introduction

Necrotizing enterocolitis (NEC) is a gastrointestinal disease affecting preterm newborns. With a 5–16% occurrence in very-low-birth-weight (VLBW) infants [1,2,3,4] and a mortality rate of 20 to 50% [5,6,7], NEC is one of the most life-threatening conditions for premature infants in neonatal intensive care units (NICU). Even in the survivors, NEC is responsible for various long-term clinical sequels [7,8,9,10]. The etiology and pathology of NEC are clearly multifactorial but are still incompletely understood. As addressed in many seminal reviews, the main risk factors of NEC development include functional and immune intestinal immaturity and dysbiosis, formula feeding and low birth weight [4,11,12,13,14,15,16,17].

NEC prediction remains a challenge in the NICU because its diagnosis mostly relies on clinical and basic radiologic characteristics [18] which are associated with features of other neonatal conditions [13]. Considering these limitations, most of the efforts for improving the prediction of NEC development in the NICU have focused on the search for molecular biomarkers. In addition to being able to distinguish NEC from other gastrointestinal conditions prior to clinical presentation [19,20,21,22], the ideal biomarker candidates should be tested from samples obtained under noninvasive conditions [17,19,21,22,23,24]. It is noteworthy that the detection of NEC before its progression to an advanced stage would allow for preventive intervention such as antibiotic treatment and feeding with a human milk diet [11,13,19,25,26,27].

The search and discovery of disease molecular targets for prognostic and diagnostic purposes is a key role in biomarker analysis studies [28]. Only a few studies have performed biomarker research on samples obtained before the onset of NEC clinical manifestations in VLBW infants. In one recent study, a large panel of biomarkers was tested in blood samples obtained from VLBW infants, but none of them identified the newborns at risk of developing NEC [29]. It has also been found that the urinary intestinal-fatty acid binding protein can predict NEC one day before clinical manifestations [30], while a combination of urinary biomarkers was shown to predict disease severity within the first 6 h of NEC suspicion [31]. The use of stool microbiome features in combination with clinical metadata features has been shown to identify NEC-affected infants more than 24 h before disease onset [32]. Specific epigenetic changes (*TLR4* gene methylation) [33] and significant fluctuation in total bile acid levels [34] were also observed in stool samples prior to NEC diagnosis. Finally, our group recently investigated a series of biomarkers in the stools of VLBW infants collected prospectively [35]. Among the tested biomarkers—which included fecal calprotectin, lysozyme, haptoglobin and intestinal alkaline phosphatase, which were previously found to be indicative of NEC near diagnosis time [36,37,38,39] and lipocalin-2, a key marker associated with the mucosal samples of NEC infants [40]—only the calprotectin and lipocalin-2 combination was found to provide a predictive signature for NEC development up to a week in advance of the diagnosis.

In the present study, we have explored SWATH mass spectrometry-based proteomics to further investigate the potential of fecal biomarkers for predicting NEC development in stool samples collected from VLBW infants in the context of a multi-center prospective study [35].

## 2. Results

### 2.1. Discovery of Up- and Down-Regulated Proteins in Stools of NEC Infants by SWATH-MS Analysis

As summarized in the flow chart (Figure 1), stool samples harvested from the seven VLBW babies over the 10-day period preceding the diagnosis of NEC (identified as day 0), as well as those of the 14 matched non-NEC VLBW infants, were pooled under three sets according to the period preceding the diagnosis. Set A was a pool of the samples obtained 10 to 7 days before diagnosis for each infant, set B was for the samples obtained 6 to 3 days before diagnosis and set C was for the samples obtained between 2 days before diagnosis and the day of the diagnosis.

Two distinct spectral libraries were generated for comparison. The first one was consolidated from the FragPipe analyses of DDA and SWATH data at 5% FDR comprising 8957 peptides (10,029 precursors) and 1044 unique proteins. The second is the reference spectral library built with DIA-NN using the library-free search mode at 1% FDR and the FASTA file of protein targets identified from the FragPipe analyses, comprising 12646 peptides (15,404 precursors) and 1175 protein groups composed of 1046 unique proteins. As expected, most of the unique proteins (Figure 2A) and peptides (Figure 2B) were shared between the two libraries, although DIA-NN identified more peptides at 1% FDR than FragPipe at 5% FDR. To confirm the validity of the DIA-NN library-free approach, the two libraries were tested with DIA-NN for the generation of quantified results.

Using the same parameter for the analysis, both libraries gave comparable results; however, more protein groups were quantified in total (1085 DIA-NN vs. 1022 FragPipe; 957 shared) and with less than 10% missing values (489 DIA-NN vs. 451 FragPipe; 416 shared; Figure 2C) with the reference spectral library built from the DIA-NN library-free search.

Starting with the 489 protein groups with less than 10% missing values from the reference spectral library, potential protein markers were selected with an area under the curve (AUC) of receiving operating characteristic (ROC) curves of 0.7 and above. Among these, 80 (69 protein groups of a single protein gene; 11 of two or more protein genes; 86 human genes in total)**,** which were also filling the other selection criteria, as defined in M&M, were selected, and half of them were found to be increased in the stools of NEC infants, the other half being reduced (Figure 3).

The list of these protein markers along with precursor information is provided in Appendix A. The principal functions of these selected protein markers were assessed by gene ontology analysis of the biological process (GO_BiologicalProcess-EBI-UniProt-GOA-ACAP-ARAP) using the ClueGO app [41]. In total, 125 genes picked from 69 of the 86 human genes were clustered between eight main groups of related biological process-enriched terms, represented by their most significant term, which were: humoral immune response (32% of the total clustered genes), acute inflammatory response (21%), proteolysis (18%) and antioxidant activity (10%), as well as tight junction assembly (5%) and digestive process (endopeptidase inhibitor activity, carbohydrate process and digestive system process for 6%, 5% and 4%, respectively) (Figure 4 and Appendix A).

### 2.2. Optimization of SWATH-MS-Derived Stool Signatures for NEC Infants

One objective of this study was to generate stool proteomic signatures that could best predict NEC development. We first chose to use principal component analysis (PCA) [42,43] to explore the discriminating information provided by the 80 potential protein markers and to evaluate sub-selections of this marker set. This multivariate unsupervised dimension reduction method was used to evaluate the capacity of the series of protein markers to cluster the samples adequately based on their outcomes. For the exploratory purpose, only the first two principal components (PC), which summarize the most variability information contained in the original dataset, were selected. As shown in Figure 5, the PC score plot illustrated the relationship between the NEC and non-NEC control samples (Figure 5A), while the protein loading plot (Figure 5B) showed how each protein marker correlated with the score plots. On the loading plot, the clear parting along the PC1 axis of increased (1 to 40) and decreased (41 to 80) markers in NEC indicates that the variation in abundance between these two sets contributes strongly to the divergent distribution of the two types of samples on the same axis as the score plots (Figure 5A). Like the markers (Figure 5B), the sample distribution on the PC2 axis formed more of a continuum (Figure 5A), but, interestingly, more matched pairs (e.g., N2A and C2A) were found on the same side of the axis (13 pairs) than on the opposite side (8 pairs), suggesting that individual characteristics have an impact on marker levels, which could explain at least one part of the variability summarized by the PC2 on both plots. The first two PCs accounted for 33.6% of the total variance of the dataset (Table 1); however, sample separation into discrete clusters was visibly incomplete (Figure 5A), as confirmed by a PC regression of PC score coordinates (PC1 and PC2) [44] that misclassified the membership of two samples (data not shown). Interestingly, parallel analysis [45] generated by Prism to correct for sampling variability and noise selected the first six PCs (Table 1) as those that represent actual variance (eigenvalues greater than the 95th percentile) in the dataset. 

To increase the variance explained by the first PCs, we reduced the number of predictor variables using ROC curve information to guide the sub-selection of complementary marker series that can predict the actual outcomes of NEC and non-NEC samples. As expected, higher cumulative proportions of variance, explained by PC1 and PC2, were found for the best 36 (18 up and 18 down; 37.1%), 20 (10 up, 10 down; 41.8%) and 14 (7 up, 7 down; 47.2%) series of protein markers (Table 1), while parallel analysis selected the first 4 PCs of the 36 series and the first PC of the 20 and 14 series (Table 1). Accordingly, the PC score plots of the best 36 (Figure 6A), 20 (Figure 6B) and 14 (Figure 6C) series show a complete separation of NEC and non-NEC sample clusters, as confirmed by PC regression (Table 2).

To further assess the 36, 20 and 14 series’ capacity to discriminate the NEC and non-NEC control samples, we tested two other classical unsupervised clustering methods (hierarchical clustering and K-means clustering) and one supervised classification method (linear discriminant analysis [42,43]) directly on the normalized scores of the expression values. Hierarchical cluster analysis was performed using the farthest neighbor method with the Pearson correlation interval. Discrete clustering was achieved with the 36 and 20 series, whereas the 14 series also formed a two-main-clusters solution, but four samples were misclustered (Table 3). K-means clustering analysis using a two-clusters solution gave similar results. Although none of the series provided complete discrete clustering, the 36 series was the more accurate one, with one sample attributed to the wrong cluster, while the 20 series had two and the 14 series had four misclustered samples (Table 3). Linear discriminant analysis is a multivariate classification method that aims to find an optimal linear function from variables that best explain predetermined classes [42,43]. This method succeeded in correctly classifying all the samples with the 36 series but not with the 20 and the 14 series, which misclassified one (Table 3). The separation of the group centroids (Chi-square test) was significant for the three series, but the lower Wilk’s Lambda value [46] of the 36 series indicated a better ability of the discriminant function to separate the cases into groups compared to the other two series.

Understandably, the counts of samples misclustered or misclassified for the three periods (A, B and C) through these three methods indicate that those from group A are more difficult to differentiate (Table 2 and Table 3). However, overall, the results show that it is feasible to discriminate between all the NEC and non-NEC samples. The signature of the 36 series showed the most efficient discrimination capacity, emphasizing the importance of selecting the right number of complementary markers to optimize the sorting of the samples. The list of these 36 selected protein markers is provided in Table 4.

### 2.3. SWATH-MS-Derived Stool Signature for Predicting NEC Development

As shown in Figure 7, and as described in the previous section, the discrete clustering of NEC (N1 to N7 group A, B, C: in red) and non-NEC samples (C1 to C7 group A, B, C: in blue) on the dendrogram of the 36 series confirms its capacity to discriminate them according to their respective type. This analysis formed relatively well-defined subclusters founded on individual cases (e.g., N5A, N5B and N5C), which seem more ostensible for the NEC samples, supporting that the sample of the same type also shows variability in association with individual characteristics. The heat map linked to the dendrogram (Figure 7) showed normalized scores of protein levels using the Rankit proportion estimation formula. The x-axis displays the 36 protein markers with the corresponding number from Table 4 (in brackets), and the y-axis displays the 42 samples in the order determined by the analysis. This heat map clearly shows the distinct profiles of the increased and decreased markers and provides a good illustration of how complementarity between and within the group of markers contributes to the discrete clustering.

The relative score of the samples for the 36 stool protein markers was evaluated as a potential tool to predict NEC development at a distinct time before the diagnosis for each subject pair. Relative scores were determined by calculating the means of the standardized values (z-score) of each sample for the 18 increased and 18 decreased protein markers separately, converting the sign of the decreased mean values (multiply by −1) and adding them to the corresponding increased mean values. As shown in Figure 8, the signature from the combination of the 36 stool proteins discriminated all seven NEC from non-NEC infants at least a week in advance. To further test the stool protein signature as a predicting tool for all VLWB infants born between 24 and 29 weeks, the seven NEC and their corresponding non-NEC controls were pooled. As shown in Figure 9, their relative expression scores were found to be significantly different from the control group for both groups of 18 proteins at all stages, as well as in combination, indicating that the 36 protein-based signatures can predict NEC development more than a week in advance for all seven tested cases.

## 3. Discussion

In this study, we have explored the power of SWATH mass spectrometry-based proteomics to investigate the potential of predicting NEC development from the stools of premature infants. Major efforts toward the discovery of predictive biochemical and clinical markers for this devastating disease have been made over the last decade, but, as mentioned above, they have had limited success in terms of days of predictability prior to the onset of clinical manifestations. This information is a key factor for allowing for an efficient preventive intervention in the NICU for NEC patients [17,47] and for avoiding the overtreatment of non-NEC infants [47,48].

Stool samples present many advantages for investigating NEC-related biomarkers. First, they can be harvested in a completely non-invasive way, even for VLBW, without the significant additional burden for NICU personnel. Second, they are likely to directly reflect the intestinal alterations that precede NEC development, as illustrated by the mucosal release and fecal accumulation of calprotectin and lipocalin2 [35], methylated TLR4 [33], microRNAs [49] and the fluctuation of fecal bile acid levels [34]. Although bowel movements may be delayed in some VLBW infants, most studies were able to successfully deal with this issue for their stool analyses [50,51,52], although, in one study, the low recovery of stool specimens at the time of the NEC acute phase was reported to reduce the clinical value of the test for NEC diagnosis [49]. In agreement with the preceding studies, herein, we have obtained a sufficient number of stool samples for each of the three selected periods to test seven of the eight NEC cases up to 10 days before the diagnosis [35].

There are only a few studies so far that have used mass spectrometry-based proteomics for discovering candidate biomarkers for NEC using samples obtained from blood [53,54], saliva [55], urine [56] or tissues [57], which have led to the identification of proteins displaying accurate diagnostic and prognostic information, but only few of them have conducted investigation before the suspicion of NEC [17]. The only one was a pilot study where proteomics was used to identify predictive biomarkers for NEC from buccal swabs [55]. To our knowledge, this is the first analysis of stool proteins procured from premature infants. Our strategy was to use SWATH-MS in the DIA mode to combine deep proteome coverage with quantitative consistency and accuracy [58]. The strategy for shotgun proteomics was adapted for the complexity of the protein/peptide matrix of the stool. Firstly, DDA and DIA data analysis with FragPipe ensured the completeness of the protein targets that were analyzed with DIA-NN in the library-free search mode. Secondly, the generation of two independent libraries from two software based on different algorithms allowed us to validate the composition of our reference spectral library and of the quantified results. It is worth noting that the latest version of FragPipe integrates DIA quantification with DIA-NN. Indeed, both the FragPipe/msFragger library and DIA-NN library-free mode have been reported to perform well in different aspects of label-free DIA signal extraction and quantification (e.g., the number of identifications, reproducibility) with DIA-NN [59,60]. Of note, the library-free mode that we used gave a higher number of spectral identifications and quantified proteins, even with the FDR estimate considered to be conservative [61,62].

Overall, 80 proteins with an average of 15 peptides (*n* = 1–100) were found to be consistently differentially detected in the stools of NEC infants relative to those of non-NEC infants. Collectively, the gene ontology analysis of the significantly differentially expressed proteins revealed that more than 60% of them were associated with immune and inflammatory responses and antioxidant defense, an observation consistent with previous observations of NEC intestinal samples analyzed by RNA-Seq, where 60% of the significant pathways were also identified in samples of patients with Crohn’s disease, including gene families related to immunity, infection, antioxidants and antimicrobials [40,63]. That 18.4% of the proteins in the proteolysis categories were observed in the NEC stool samples is also interesting, since fecal proteolytic and elastase activity were reported to be increased before the onset of ulcerative colitis [64]. Interestingly, the increase in fecal proteolytic activity was associated with gut microbiota changes in patients with ulcerative colitis [64], while only human sequences were considered in this work. In this context, it is pertinent to note that a number of markers involved in proteolysis were pancreatic enzymes such as CELA2A, CELA3A, CELA3B and CTRC, which were all found to be reduced in the stools of NEC infants, an observation consistent with the general immaturity of the gut in premature infants who are most susceptible to developing NEC, which may also include limited pancreatic functions [65].

On the other hand, this study also aimed to evaluate the series of markers needed to find the best signature of NEC. While the evaluation of the 36, 20 and 14 series was founded on different unsupervised clustering and supervised classification analyses, the 36 series was able to discriminate more samples on average and showed a lower (better) Wilk’s Lambda value in linear discriminant analysis. Although the variation of abundance between increased and decreased markers seems central to sorting the sample according to their actual outcome, the PCA and hierarchical clustering analysis results indicated that some level of variability within these sets of markers explains the variability between the samples of the same type. In this sense, reducing the number of markers below a certain threshold may limit the primary and complementary information available to sort the samples, which could explain why the 36 series performed better than the 20 and 14 series. However, as the number of samples is limited in this study, it is essential to mention that we do not consider this specific selection of protein markers to be the absolute NEC signature. Indeed, similar work on larger cohorts will be needed to establish more robust conclusions.

It is pertinent to note that, of the 36 selected proteins, 18 were found to be increased in the stools of NEC infants, while the other 18 were reduced. The findings for some of the stool proteins appear consistent with expectations based on the literature, such as the increase in fecal S100A12 in NEC infants [66] and the elevation of mucosal DEFA5 [40,63], Serpin B1 [67] and peroxiredoxin-1 [57] in NEC infants and/or patients with inflammatory bowel disease. Stool accumulation of brush border enzymes such as trehalase and sucrase-isomaltase is also consistent with the extensive destruction of the small intestinal epithelium noted in NEC infants [57]. However, the reduction in the fecal levels for claudin-3 and SLC26A3 observed herein is more puzzling, since claudin-3 was reported to increase in the urine of neonates with NEC [68], and the expression of SLC26A3, an intestinal brush border chloride anion, is drastically reduced in patients with inflammatory bowel disease and has recently been shown to regulate epithelial barrier integrity [69]. Since these two proteins are reduced in the stools of infants more than a week before NEC is diagnosed, one may speculate that the lower fecal levels of these paracellular components could be indicative of a potential mucosal defect in NEC infants.

Another issue that needs to be pointed out is the absence of the calprotectin S100A8 and 100A9 components and lipocalin-2 (LCN2) in the final list of selected proteins, considering that they were previously identified as a robust stool biomarker combination for predicting more than half of the NEC infants [35]. One key aspect of the current study to consider was that each biomarker was selected on the basis of its ability to discriminate NEC vs. non-NEC infants over the 10-day period preceding diagnosis. Indeed, as observed for calprotectin and LCN2 in the immunoassay [35], approximately half of the S100A8, S100A9 and LCN2 in the NEC samples displayed an overlap with the non-NEC samples. S100A8, S100A9 and LCN2 were therefore not retained based on the AUC of ROC analysis. This is consistent with the fact that our approach favored the robustness of the analysis, in line with the current biomarker discovery strategies [70]. Indeed, the accurate identification and quantification of the proteins (ensured here with the hybrid DDA/DIA strategy and by inferring protein libraries from two independent software) is paramount to biomarker discovery and outweighs the identification of as many proteins as possible [71] while reducing the likelihood of false positives [72].

The optimized signatures based on the selection of the 36 proteins found to be increased and decreased in the stools of the NEC infants were able to predict NEC development in all seven VLWB infants at least one week before the diagnosis. This finding is a major achievement in the context of a non-invasive diagnosis method considering that previous published detection methods were predicting either much less in advance [30,31,32,33] or with less sensitivity [34,35].

It is, however, important to consider this analysis as a proof-of-concept study demonstrating that label-free quantification using SWATH/DIA mass spectrometry-based proteomics assisted with DIA-NN software can successfully be applied to stool analyses for the prediction of NEC development. Indeed, despite being derived from a multi-center prospective study where the stools from 132 VLBW infants were collected, NEC is a relatively infrequent condition occurring in the NICU, so only eight of them received a stage-3 diagnosis, including one for which no stools were available for the 7–10 day period before the diagnosis. In this context, considering the relatively small size of the NEC cohort, the set of the 36 protein markers selected for the establishment of an “NEC signature” should not be seen as definitive, since some variations in the levels of proteins may not be related to the disease but rather are a first step toward the usual strategy for mass spectrometry-based proteomics biomarker discovery, where a small number of samples are analyzed in-depth to select the markers to be tested in the next phases [70,73]. The procedure described herein for the selection of the protein markers should therefore be useful for refining this preliminary NEC signature in a future study with a larger cohort required prior to the design of a validation phase for the selected biomarkers by using targeted proteomics analysis. Another aspect that should be considered in a future study would be the inclusion of samples from premature infants affected with other gut-related diseases to optimize an NEC-specific signature.

## 4. Materials and Methods

### 4.1. Sample Collection and Preparation for LC-MS/MS Analysis

Stools from 132 very-low-birth-weight infants were collected on a daily basis in the context of a multi-center prospective study aimed at investigating the potential of fecal biomarkers for NEC prediction [35]. Eight infants received a stage-3 NEC diagnosis. The stools collected up to 10 days before the diagnosis were available for seven of them. They were matched with seven pairs of non-NEC controls and grouped according to three test periods (group A: −10 to −7, group B: −6 to −3 and group C: −2 to 0 days before diagnosis), as described before [35], for a total of 42 unique samples. For the preparation, 100 mg of each stool specimen was solubilized in 1 mL of Tris buffer (25 mM Tris pH 7.5), pooled in groups (as detailed above) for NEC and non-NEC infants and centrifuged (16,000× *g*; 4 °C) for 30 min. The aqueous phase between the pellet and the floating residuals was recovered and stored at −80 °C until preparation for LC-MS/MS analysis.

The concentration of solubilized proteins in the individual samples was measured using a BCA protein assay (ab102536, Abcam, Cambridge, UK). For the 42 individual sample preparations, volumes corresponding to 40 μg of protein were brought up to 100 μL in a 50 mM Tris buffer pH 8.0 for reduction (dithiothreitol (DTT): 10 mM, 10 min at 65 °C), alkylation (15 mM iodoacetamide, 30 min in the dark at room temperature) and quenching (10 mM DTT) steps. The proteins were recovered by precipitation in acetone for 1 h at −80 °C, followed by centrifugation at 16,000× *g* for 15 min, and the protein pellets were washed with cold methanol. The enzymatic digestion of proteins was carried out in 100 μL of 50 mM Tris buffer pH 8 using 0.7 μg per sample of the proteolytic mix Trypsin/Lys-C, MS Grade (Promega) for 2 h at 37 °C, continued overnight with an additional 0.7 µg of the proteolytic mix (30:1 *w*/*w* protein/protease total ratio) and stopped with 2% formic acid. The cleaning and recovery of the peptides were done with a reverse-phase Strata-X polymeric SPE sorbent column (Phenomenex) according to the manufacturer’s instructions. The recovered peptides were dried under nitrogen flow at 37 °C for 45 min and stored at 4 °C until being resuspended in 20 µL of mobile phase solvent A (see the section below) before LC-MS/MS analysis. To generate an ion library, extracted proteins from a representative pool of samples (three NEC and two non-NEC, covering the three test periods) were separated on a 4–20% polyacrylamide gel. The proteins were separated in 12 fractions and then reduced, alkylated and digested in the gel. Peptides were extracted from the gel by successive rounds of dehydration and sonication and purified using reverse-phase SPE.

### 4.2. LC-MS/MS Acquisition

LC-MS/MS data acquisitions were carried out at the proteomics facilities of PhenoSwitch Bioscience Inc. (Sherbrooke, QC, Canada). The acquisitions were conducted with an ABSciex TripleTOF 6600 (ABSciex, Foster City, CA, USA) equipped with an electrospray interface with a 25 μm iD capillary and coupled to an Eksigent μUHPLC (Eksigent, Redwood City, CA, USA). Analyst TF 1.8 software was used to control the instrument and for data processing and acquisition. The acquisition was performed in the Data-Dependent Acquisition (DDA) mode for the 12 fractions of the ion library. The 42 individual samples (10 µg) were analyzed in the Sequential Window Acquisition of All Theoretical Mass Spectra (SWATH) acquisition mode [74]. The source voltage was set to 5.5 kV and maintained at 325 °C, the curtain gas was set at 35 psi, gas one was set at 27 psi and gas two was set at 10 psi. Separation was performed on a reverse-phase Kinetex XB column 0.3 mm i.d., 2.6 μm particles, 150 mm (Phenomenex), which was maintained at 60 °C. Samples were injected by loop overfilling into a 5 μL loop. For the 60 min LC gradient, the mobile phase consisted of the following: solvent A (0.2% *v*/*v* formic acid and 3% DMSO *v*/*v* in water) and solvent B (0.2% *v*/*v* formic acid and 3% DMSO in EtOH) at a flow rate of 3 μL/min.

### 4.3. MS Data Preparation and Pre-Analysis

For label-free quantification, SWATH/DIA is reported to offer better proteome coverage, reproducibility and quantitative precision compared to DDA [75]. The DIA-NN software [62] can reproduce protein quantification obtained by the semi-automatic but time-consuming curation of precursor spectra and proteins with the software Skyline (MacCoss Lab, University of Washington, Seattle, WA, USA) [76]. However, the computer calculation time from the library-free search mode with DIA-NN increases rapidly with the size of the reference fasta file and the dataset. Trying to build a library from the whole human proteome is not optimal. For this reason, and considering the published workflow on different MS settings and instruments comparing software performance and using cross-compatible workflow [59,60,61], we opted for a hybrid strategy combining the fast and robust FragPipe Graphical User Interface (GUI; v.17.1; https://fragpipe.nesvilab.org/; accessed on 15 March 2022) proteomics platform (integrate MSFragger, Philosopher and EasyPQP) [77,78,79] for pre-analysis of the data and for creating a list of protein targets from preliminary libraries and DIA-NN (GUI; v.1.8) for generating the reference spectral library from this list and for the quantification of the proteins.

To produce an MS data format compatible with FragPipe and DIA-NN, Sciex DDA and SWATH, .WIFF files were converted to .mzML format with MSConvert (GUI) from ProteoWizard (v3.0.22074) [80] using the Peak Picking filter with the “Vendor” algorithm. The .mzML from SWATH data were further converted to pseudo-MS/MS spectra .mzML files using DIA-Umpire [81] in MSConvert. From FragPipe, MSFragger (v.3.4) was used to perform proteomic searches against the human proteome reviewed database (UP000005640; isoforms and contaminants included; www.uniprot.org; accessed on 15 March 2022), with mostly default open search settings (peptide length from 6 to 42, enzyme set to stricttrypsin, missed cleavage set to 1, max fragment charge set to 4, methionine oxidation set as variable modification and carbamidomethylation set as fixed modification) [82,83]. Philosopher was used (v4.1.1) for the proteomic validation (peptide spectrum matches (PSM) validation, protein inference and false discovery rate [FDR] filtering), and EasyPQP (https://github.com/grosenberger/easypqp; accessed on 15 March 2022) was used for spectral library generation from the filtered results.

To maximize the extent of the list of protein targets, three preliminary spectral libraries were built from FragPipe using a permissive 5% FDR to filter the results: one from the DDA data, one from the SWATH pseudo-MS/MS data and one from the DDA and SWATH pseudo-MS/MS data, concurrently. These libraries were combined and filtered to remove the duplicates and counted a total of 1087 protein UniProtKB IDs entries and 1044 unique proteins (isoform IDs of the same protein counted as one protein). Two proteins of interest were added to this list (alkaline phosphatase, germ cell type [ALPG], and bovine beta-lactoglobulin [LBG]) and the “retrieve/ID mapping tool” from Uniprot.org (https://www.uniprot.org/uploadlists/; accessed on 16 March 2022) has been used to create a FASTA file containing the 1089 protein targets and their amino acid sequence.

### 4.4. Reference Spectral Library Building and Label-Free Quantification Analysis

The reference spectral library was built with DIA-NN using the 1089 protein target database .fasta file as reference. First, an in-silico-predicted spectral library was created from the sequence database with the FASTA digest library-free search/generation and deep learning based-prediction mode enabled. This library was used to analyze the 42 individual .mzML files of the SWATH runs, keeping equivalent settings for precursor length, missed cleavage, charge and modifications, as described above. Match between runs (MBR) was enabled to create a new library and reanalyze the data from this library (precursor FDR and matrix q value FDR set to 1%). The “remove likely interference” and “reannotate” options were activated, and the human proteome reviewed database was used as the reference .fasta file, with protein inference set to genes. The DIA-NN protein inference algorithm (set to genes) creates protein groups according to the gene from which the precursor(s) can be annotated in the reference .fasta file. Precursors from proteotypic peptides are attributed to single protein groups that correspond to a single gene, whereas indistinguishable non-proteotypic precursors are also merged to protein groups that combine the single proteins from which they can originate. The DIA-NN reference spectral library counted 1174 protein groups from 1046 unique protein names, of which 975 protein groups were composed of a single protein, and 199 were composed of two proteins or more. A total of 1085 protein groups were found to have quantified results in the protein groups (pg)_matrix output file, which contains the normalized expression values calculated by DIA-NN. The label-free quantification method applied is based on the MaxLFQ algorithm [84], using only the optimal ion fragments from the top precursors passing the run-specific and global q value threshold (1% FDR) [62].

The mass spectrometry proteomics data have been deposited to the ProteomeXchange Consortium [85] via the PRIDE partner repository [86] with the dataset identifier PXD036178.

### 4.5. Dataset Composition, Protein Marker Selection and Series Evaluation

From the pg_matrix file, only protein groups with less than 10% missing values were kept for the statistical analysis, for a total of 489 protein groups with ≥38 quantified values on the 42 samples. Since the missing values in the pg_matrix indicate the absence or very low abundance of precursors, zero imputation was applied to missing values. Then, receiver operating characteristic (ROC) curve analysis was performed to assess the binary discriminant value of individual protein groups, and those with an area under the curve (AUC) ≥ 0.7 were kept. To ensure the selection of high-quality and non-biased features, this list of potential protein markers was further refined by eliminating the protein groups with four imputed zero values applied to one particular type of sample (NEC or non-NEC control) and those lacking a precursor with at least 35 quantified values in the precursor (pr)_matrix output file. A maximum of two protein groups containing redundant entries were kept as distinct protein markers. The final list of potential markers thus counted 80 protein groups, 40 with levels increasing in NEC, and 40 with levels decreasing in NEC. Starting with these 80 protein groups, selection for the series of 36, 20 and 14 protein markers was executed successively from the precedent, keeping the protein groups with the best individual AUC, and the others were selected based on their impact on communal ROC curves (increased and decreased taken separately) that collectively show a greater capability to predict the outcomes of NEC and non-NEC samples.

For each series selection, multidimensional analysis was performed using principal component (PC) analysis to observe the behavior of the individual samples and the variables (protein group levels) on the two main dimensions (PC1 and PC2) [87]. PC regressions were based on binary logistic regression of the PC scores from the first two PCs as independent variables and the two a priori group knowledge (NEC and non-NEC control) as dependent variables [44]. Hierarchical cluster analyses were performed with the farthest neighbor (complete linkage based on the similarity of the farthest pair) method and Pearson’s correlation intervals (comparing two vectors of values) [88]. The PC regressions K-means clustering analysis [89] and linear discriminant analysis [90] were used to assess the feasibility of the classification and clustering models from the protein marker series selection. However, no cross-validation methods were tested, since this was only for exploratory purposes, and no validation cohort was available.

### 4.6. Statistical Analysis and Graph

All calculations and graph plotting were carried out with GraphPad Prism (Prism v.9.3.1; GraphPad Software, Inc) or IBM SPSS Statistics (SPSS v28.0.1.1; IBM), except for the Venn diagrams, which were created with the web-based tool Venny (v.2.1) [91], and the Gene Ontology (GO) analysis, which was conducted with the ClueGO (v.2.5.8) plug-in from the Cytoscape software (v.3.9.1) [41,92].

## 5. Conclusions

This proof-of-concept study demonstrates that SWATH/DIA mass spectrometry-based proteomics used for analyzing the stools of premature infants represent a promising potential approach for predicting NEC development in NICU. The next steps should include the refinement of this NEC signature in a future study with a larger cohort that should include premature infants affected by non-NEC intestinal diseases and a validation step using targeted proteomics analysis.

## Figures and Tables

**Figure 1 ijms-23-11601-f001:**
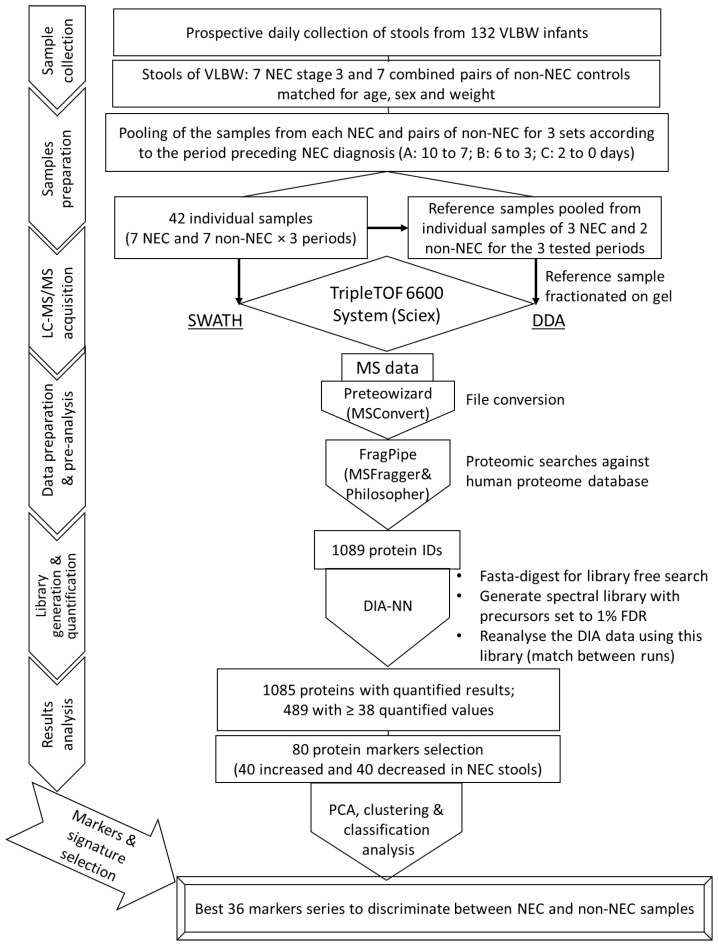
Flowchart illustrating the main steps of this study.

**Figure 2 ijms-23-11601-f002:**
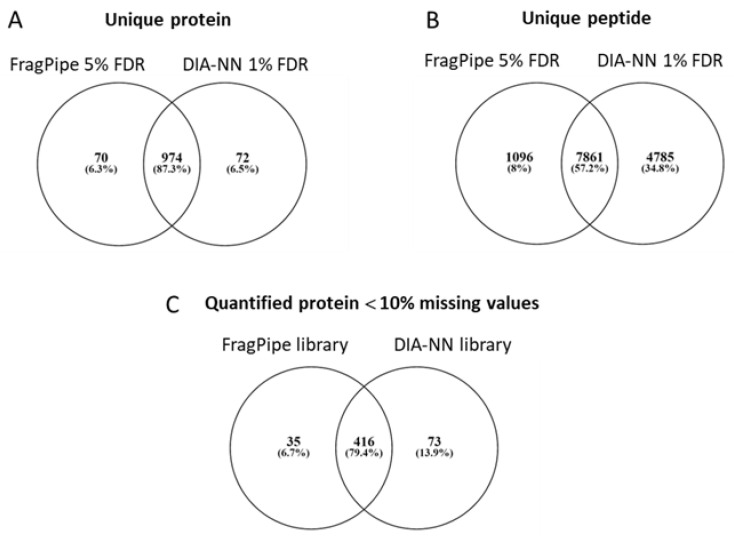
Comparison of libraries and quantified results from FragPipe and DIA-NN analysis. (**A**,**B**) Venn diagram comparing unique proteins (**A**) and unique peptides (modified and unmodified; (**B**)) identified in the two libraries. (**C**) Venn diagram comparing protein groups with less than 10% missing values quantified with DIA-NN using the combined FragPipe library or DIA-NN library-free mode. FDR: false discovery rate.

**Figure 3 ijms-23-11601-f003:**
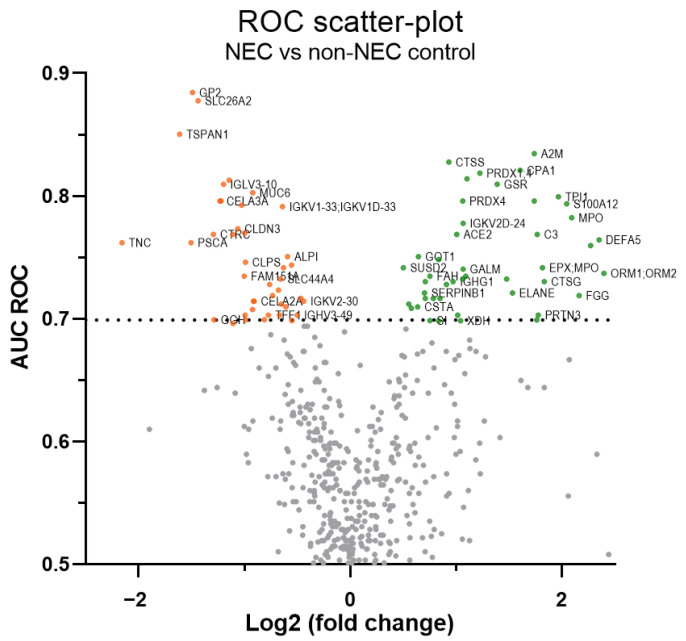
ROC scatterplot summarizing differentially abundant proteins in NEC stools. “Volcano-type” ROC scatter-plot of 489 proteins with ≥38 quantified values. On the plot, the x-axis represents the log2 fold change ratio of the mean protein levels of NEC vs. non-NEC control samples, the y-axis is the area under the curve (AUC) of the receiver operating characteristic (ROC) curve and the orange and green dots show down- and up-regulated protein levels passing the 0.7 AUC threshold (dashed line), respectively. For more clarity, gene symbols are used on the plot (see Appendix A for corresponding protein name). Some labels have been skipped.

**Figure 4 ijms-23-11601-f004:**
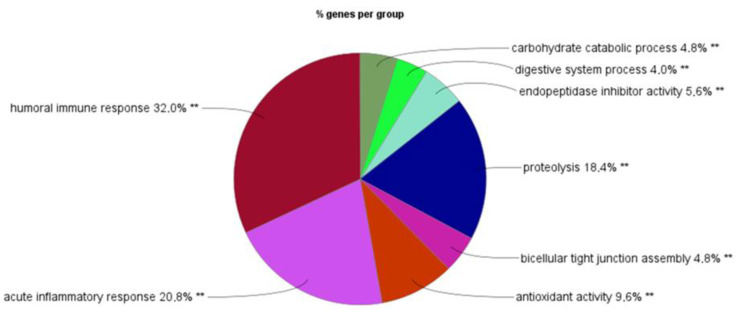
Gene ontology of the biological process. Gene ontology analysis of the 86 human genes associated with the 80 protein markers using the GO_BiologicalProcess-EBI-UniProt-GOA-ACAP-ARAP (22–04-2022) reference term database with the ClueGO (v.2.5.8) plug-in in the Cytoscape software (v.3.9.1). In total, 125 genes picked from 69 of the 86 genes were clustered between the eight main groups of biological process-related terms. On the pie chart, the groups are named based on their leading (most significant) term, and the size of the segment is defined by the % of associated genes per group calculated from the total picked genes. Group-corrected *p* value calculated by ClueGO with the Bonferroni step-down method: ** *p* < 0.001.

**Figure 5 ijms-23-11601-f005:**
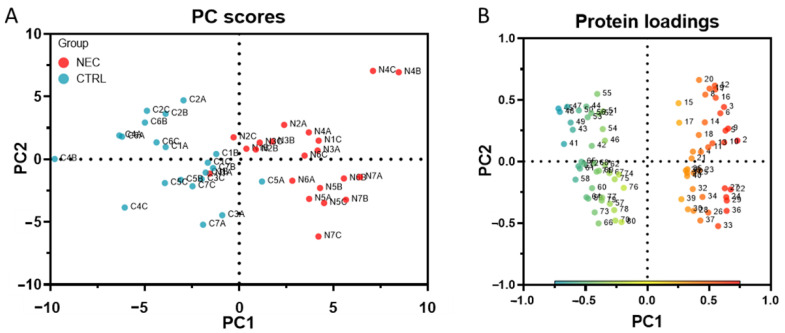
PCA of the 80 protein markers. (**A**) The PC score plots illustrate the relationship between the NEC (N1 to N7 group A, B, C: in red) and non-NEC control (C1 to C7 group A, B, C: in blue) samples when the dimensionality of the dataset is reduced to its two principal components (PC1: x-axis; PC2: y-axis). (**B**) Protein loadings show how each protein marker (increased in NEC: from 1 to 40; decreased from 41 to 80; see Appendix A for corresponding protein name) is correlated with the two principal components of the PC score plots.

**Figure 6 ijms-23-11601-f006:**
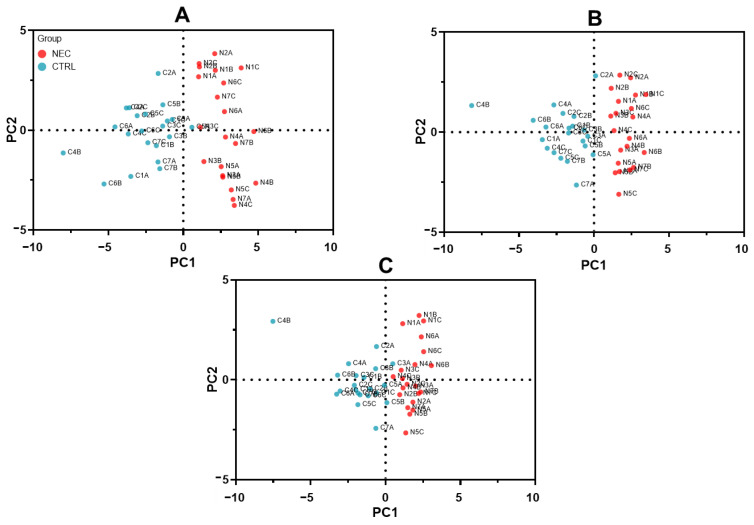
PCA of the 36, 20 and 14 series of markers. (**A**–**C**) PC score plots from the PCA of the (**A**) 36 (18 up, 18 down), (**B**) 20 (10 up, 10 down) and (**C**) 14 (7 up, 7 down) series of protein markers clustering showing the capacity of this series to form discrete clusters of the NEC (N1 to N7 group A, B, C: in red) and non-NEC control (C1 to C7 group **A**–**C**: in blue) samples with the two first principal components (PC1: x-axis; PC2: y-axis).

**Figure 7 ijms-23-11601-f007:**
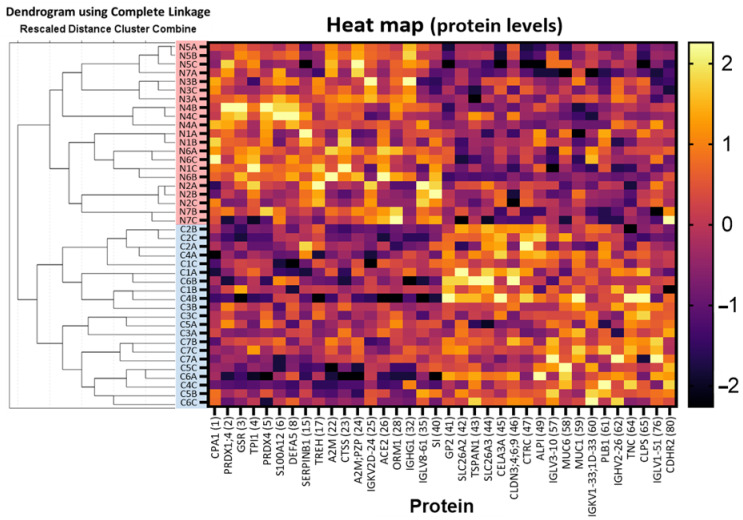
Hierarchical cluster analysis and heatmap of the 36 series. The dendrogram on the left displays the discrete clustering of NEC (N1 to N7 group A, B, C: in red) and non-NEC control samples (C1 to C7 group A, B, C: in blue) between two main clusters. The heat map linked to the dendrogram shows normalized scores of the protein levels for the 36 protein markers (X-axis) with their corresponding number (in brackets; see Table 4 for the protein name) and the 42 individual samples in the order determined by the hierarchical clustering analysis (Y-axis). The colored reference chart on the right indicates the normalized scores (range from −2.26 to 2.26).

**Figure 8 ijms-23-11601-f008:**
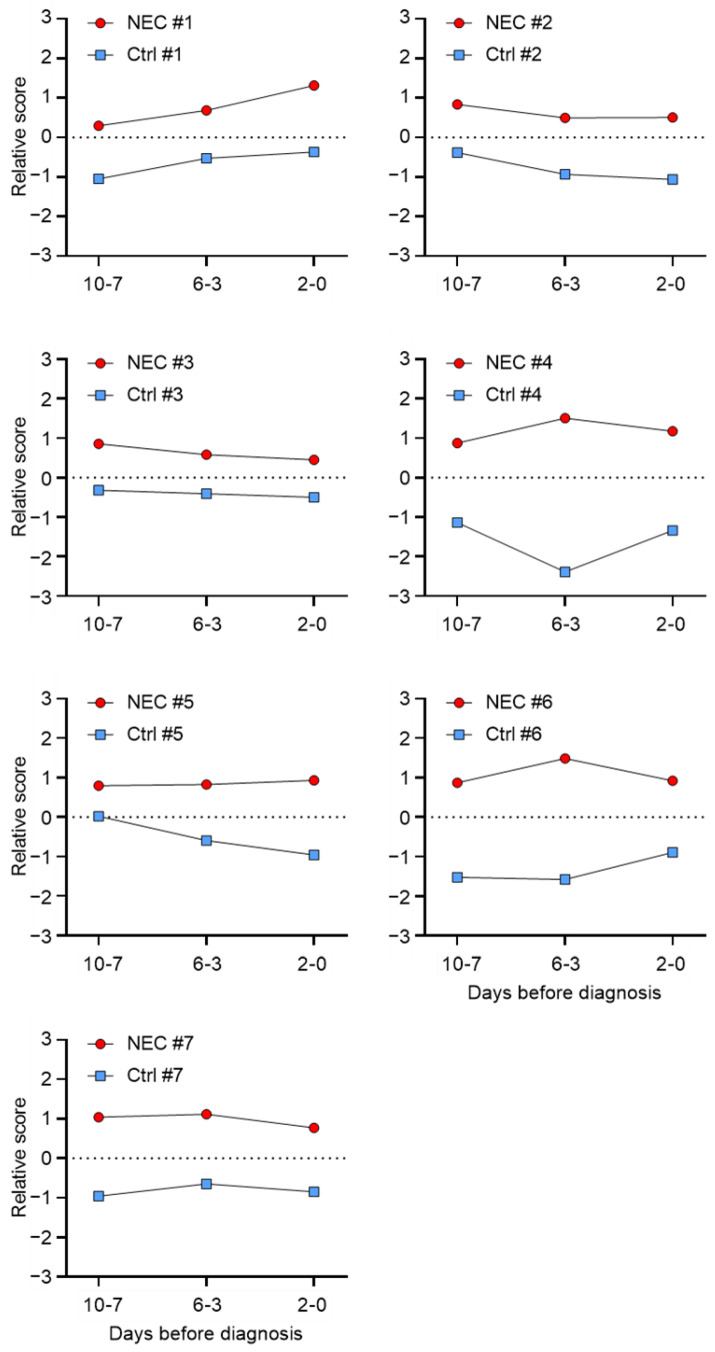
Relative score according to paired samples. Plots of the relative score (Y-axis) of the 36 protein marker levels across the three tested periods (X-axis: 10-7, 6-3, 2-0 days before NEC diagnosis) for each of the 21 paired NEC (red dot) and non-NEC control (blue square) samples from the seven paired cases. The two-tailed paired t test on the 36 protein markers (increased and decreased; multiply by −1) gave a significant difference of at least *p* ≤ 0.05 for all the pairs of the samples.

**Figure 9 ijms-23-11601-f009:**
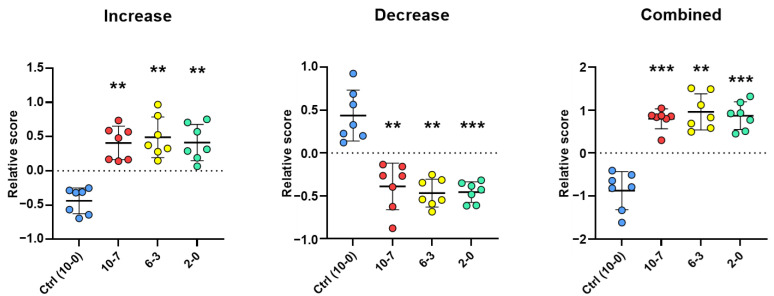
Relative score by period. The plots show the relative scores (Y-axis) for each of the seven NEC samples (red, yellow and green dots) of the three tested periods (X-axis: 10-7, 6-3, 2-0 days before NEC diagnosis) compared to the average of the three same periods for the non-NEC control samples (Ctrl: 10-0 days; blue dots) to assess the overall relative scores for samples from NEC and non-NEC infants without pairing. The (Increase) and (Decrease) plots were calculated from the 18 increased and the 18 decreased protein markers taken separately, and the (Combined) plot combines the mean values of these two plots for each sample (increased + decreased; multiply by −1). The means ±SD are shown on the plots by horizontal bars. *p*-values were calculated using one-way ANOVA Dunnett’s multiple comparisons test. ** *p* < 0.01; *** *p* < 0.001.

**Table 1 ijms-23-11601-t001:** PCA information on series’ principal components.

Series	80 Markers			
PC summary	**PC1**	**PC2**	**PC6**	**PC41**
Eigenvalue	18.23	8.64	4.12	0.04
Proportion of variance	22.79%	10.80%	5.15%	0.06%
Cumulative proportion of variance	22.79%	33.58%	62.28%	100.00%
Parallel analysis PC selection	Selected	Selected	Last selected	
**Series**	**36 markers**			
PC summary	**PC1**	**PC2**	**PC4**	**PC36**
Eigenvalue	9.31	4.05	2.68	0.002
Proportion of variance	25.86%	11.24%	7.45%	0.00005%
Cumulative proportion of variance	25.86%	37.10%	54.08%	100.00%
Parallel analysis PC selection	Selected	Selected	Last selected	
**Series**	**20 markers**			
PC summary	**PC1**	**PC2**		**PC20**
Eigenvalue	6.15	2.21		0.06
Proportion of variance	30.77%	11.05%		0.28%
Cumulative proportion of variance	30.77%	41.82%		100.00%
Parallel analysis PC selection	Selected			
**Series**	**14 markers**			
PC summary	**PC1**	**PC2**		**PC14**
Eigenvalue	4.72	1.89		0.08
Proportion of variance	33.74%	13.49%		0.60%
Cumulative proportion of variance	33.74%	47.22%		100.00%
Parallel analysis PC selection	Selected			
PC: principal component				

**Table 2 ijms-23-11601-t002:** PC regression on the first PC score coordinates.

	Samples	36 Markers	20 Markers	14 Markers
Classification	NEC (N)	21N	21N	21N
	CTRL (C)	21C	21C	21C
Misclassified by period	Gr. A	0	0	0
	Gr. B	0	0	0
	Gr. C	0	0	0

**Table 3 ijms-23-11601-t003:** Clustering and classification analysis.

Hierarchical Clustering		36 Markers	20 Markers	14 Markers
NEC (N)	Cluster 1	21N	21N	21N/4C
Ctrl (C)	Cluster 2	21C	21C	17C
Misclustered by period	Gr. A	0	0	3
	Gr. B	0	0	1
	Gr. C	0	0	0
K-means clustering		36 markers	20 markers	14 markers
NEC (N)	Cluster 1	21N/1C	21N/2C	21N/5C
Ctrl (C)	Cluster 2	20C	19C	16C
Misclustered by period	Gr. A	1	2	4
	Gr. B	0	0	1
	Gr. C	0	0	0
Linear discriminant analysis		36 markers	20 markers	14 markers
Classified	NEC (N)	21N	21N/1C	21N/1C
	Ctrl (C)	21C	20C	20C
Wilk’s Lambda		0.035	0.134	0.169
Significance (chi-square)		<0.001	<0.001	<0.001
Misclassified by period	Gr. A	0	1	1
	Gr. B	0	0	0
	Gr. C	0	0	0

**Table 4 ijms-23-11601-t004:** Selection of the 36 series of increased and decreased protein markers.

No.	First Protein Name ^1^	Protein & Protein Groups (Gene Symbol)	Quantified Values (42 Samples)	Fold Change Ratio ^2^ NEC/Control	ROC Curve AUC	Mann–Whitney Test ^3^	
						*p*-Value	*q*-Value
More abundant in NEC stools							
1	Carboxypeptidase A1	*CPA1*	42	3.04	0.82	<0.001	0.003
2	Peroxiredoxin-1	*PRDX1;PRDX4*	39	2.33	0.82	<0.001	0.003
3	Glutathione reductase, mitochondrial	*GSR*	42	2.61	0.81	<0.001	0.003
4	Triosephosphate isomerase	*TPI1*	40	3.90	0.80	0.001	0.004
5	Peroxiredoxin-4	*PRDX4*	41	2.09	0.80	0.001	0.004
6	Protein S100-A12	*S100A12*	39	4.12	0.79	0.001	0.004
8	Defensin-5	*DEFA5*	42	5.09	0.76	0.003	0.009
15	Leukocyte elastase inhibitor	*SERPINB1*	42	1.62	0.72	0.014	0.023
17	Trehalase	*TREH*	41	1.63	0.72	0.016	0.023
22	Alpha-2-macroglobulin (1)	*A2M*	42	3.33	0.83	<0.001	0.002
23	Cathepsin S	*CTSS*	42	1.91	0.83	<0.001	0.003
24	Alpha-2-macroglobulin (2)	*A2M;PZP*	42	3.33	0.80	0.001	0.004
25	Probable non-functional Ig kappa variable 2D-24	*IGKV2D-24*	42	2.09	0.78	0.002	0.007
26	Angiotensin-converting enzyme 2	*ACE2*	42	2.01	0.77	0.002	0.007
28	Alpha-1-acid glycoprotein 1 (1)	*ORM1*	42	4.82	0.76	0.003	0.009
32	Ig heavy constant gamma 1	*IGHG1*	42	1.96	0.73	0.010	0.018
35	Ig lambda variable 8-61	*IGLV8-61*	42	1.80	0.72	0.016	0.023
40	Sucrase-isomaltase, intestinal	*SI*	42	1.68	0.70	0.052	0.055
**Less abundant in NEC stools**							
41	Pancreatic secretory granule mb. major gp GP2	*GP2*	42	0.36	0.88	<0.001	<0.001
42	Sulfate transporter	*SLC26A2*	41	0.37	0.88	<0.001	<0.001
43	Tetraspanin-1	*TSPAN1*	42	0.33	0.85	<0.001	0.001
44	Chloride anion exchanger	*SLC26A3*	40	0.45	0.81	<0.001	0.003
45	Chymotrypsin-like elastase family member 3A (1)	*CELA3A*	42	0.43	0.80	0.001	0.004
46	Claudin-3	*CLDN3,4,6,9*	42	0.48	0.77	0.002	0.007
47	Chymotrypsin-C	*CTRC*	42	0.41	0.77	0.002	0.007
49	Intestinal-type alkaline phosphatase	*ALPI*	42	0.66	0.75	0.005	0.012
57	Ig lambda variable 3-10	*IGLV3-10*	42	0.44	0.81	<0.001	0.003
58	Mucin-6	*MUC6*	42	0.53	0.80	0.001	0.004
59	Mucin-1	*MUC1*	42	0.43	0.80	0.001	0.004
60	Ig kappa variable 1D-33	*IGKV1-33*	42	0.64	0.79	0.001	0.004
61	Phospholipase B1, membrane-associated	*PLB1*	38	0.49	0.79	0.001	0.004
62	Ig heavy variable 2-26	*IGHV2-26*	38	0.50	0.77	0.002	0.007
64	Tenascin	*TNC*	41	0.22	0.76	0.003	0.009
65	Colipase	*CLPS*	42	0.50	0.75	0.006	0.014
76	Ig lambda variable 1-51	*IGLV1-51*	42	0.57	0.70	0.028	0.032
80	Cadherin-related family member 2	*CDHR2*	42	0.68	0.70	0.038	0.041

Abbreviation: AUC: area under the curve; gp: glycoprotein; Ig: immunoglobulin; MMP: metalloproteinase; NEC: necrotizing enterocolitis; No.: number; ROC: receiver operating characteristic. ^1^ Duplicated first protein names come from protein groups formed by DIA-NN when indifferentiable non-proteotypic precursors can be attributed to more than one gene. ^2^ The fold change ratio was calculated from the mean expression value of both groups. ^3^ *p*-values and *q*-values have been calculated on the standardized expression values using the Mann–Whitney test for the *p*-value and Benjamini–Krieger–Yekutieli’s two-stage step-up method for the *q*-value.

## Data Availability

The mass spectrometry proteomics data have been deposited to the ProteomeXchange Consortium via the PRIDE partner repository http://www.ebi.ac.uk/pride with the dataset identifier PXD036178.

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
