# Peer review of "Proteomics Profiling of Stool Samples from Preterm Neonates with SWATH/DIA Mass Spectrometry for Predicting Necrotizing Enterocolitis"

_ijms, 2022, doi:10.3390/ijms231911601_

Round 1
Reviewer 1 Report
This manuscript describes a proteomic study of stool samples from very low birth weight (VLBW) infants, aiming to discover fecal biomarkers to predict necrotizing enterocolitis (NEC). Fecal samples were collected from infants diagnosed with stage 3 NEC, matched with pairs of non-NEC controls, and analyzed by label-free SWATH/DIA-MS approaches. Subsequently, eighty differentially abundant proteins were selected to be evaluated for their potential as biomarkers by PCA analysis, and the number of potential protein markers was further reduced to 36, 20, and 14 by maximizing the variance explained by the first principal component. Then several classification methods were performed to assess the discrimination capacity of the previously selected three series of biomarkers, and the 36 series was found to provide the best prediction power. Finally, the author developed a simple signature score calculated from the abundance level of the 36 protein biomarkers and demonstrated that it could accurately predict NEC development more than a week in advance for all seven tested cases. Together, this study lays the groundwork for identifying protein biomarkers for predicting necrotizing enterocolitis from preterm neonates. Proteomic analysis of stool samples from a larger cohort may lead to generalizable and sensitive methods for identifying infants with a high risk of NEC for early diagnosis and intervention.
Overall, this paper contains a significant body of well-thought-out and well-executed work. I recommend publication upon addressing the point detailed below.
- The author nicely demonstrated that the relative score could differentiate NEC from paired non-NEC control. It is interesting to know how effective the score is when applying it to non-NEC samples without pairing for age, sex, and weight. Could the author apply the ‘stool signature score’ to other samples from non-NEC infants in this study?
Author Response
Thank you for your positive comments and suggestions.
COMMENTS AND SUGGESTIONS:
The author nicely demonstrated that the relative score could differentiate NEC from paired non-NEC control. It is interesting to know how effective the score is when applying it to non-NEC samples without pairing for age, sex, and weight. Could the author apply the 'stool signature score' to other samples from non-NEC infants in this study?
RESPONSE:
The overall relative scores of samples from NEC and non-NEC samples (without pairing for age, sex and weight) have been calculated and are presented in Fig. 9 in p.13. For more clarity, we have added the following sentence: “(…) to assess the overall relative scores for samples from NEC and non-NEC infants without pairing.” in the legend (lines 300-301).
Reviewer 2 Report
David Gagné and co-workers in the manuscript entitled “Proteomics profiling of stool samples from preterm neonates with SWATH/DIA mass spectrometry for predicting necrotizing enterocolitis” presented the data showing proteomic analysis in Necrotizing enterocolitis infants and healthy controls.
the number of NEC studies on the proteome level are very few in general, particularly those where healthy controls are also included for comparison. Moreover, to the best of knowledge the current study describes the largest number of proteins with altered amount. Therefore, the presented study could be of high interest to readers however it should be improved first.
1. For comparison a relatively small group was used (n=7), thus authors should consider some variations in the levels of proteins which are not related to disease.
2. There is only limited discussion concerning how does the current study relate to results of previous proteomic studies in NEC. Do previous proteomic studies support the authors' conclusions?
3. In addition, a major concern hat must be addressed is related to the lack of validations performed on differentially expressed proteins by other methodologies such as western blots, ELISA, IF, IHC, even considering that these validations can be easily performed on higher number of samples respect to proteomic analyses.
Author Response
Thank you for your positive comments and suggestions.
COMMENTS:
- For comparison a relatively small group was used (n= 7), thus authors should consider some variations in the levels of proteins which are not related to disease.
RESPONSE:
We have modified the end of the discussion to consider this possibility as follows (lines 416-418): “(…) a “NEC signature” should not be seen as definitive since some variation in the levels of proteins may not be related to the disease but rather as a first step toward (…)”.
COMMENTS:
- There is only limited discussion concerning how does the current study relate to results of previous proteomic studies in NEC. Do previous proteomic studies support the authors' conclusions?
RESPONSE:
As suggested, we have further discussed previous proteomics studies in NEC as follows (lines 321-324): “(…) using samples obtained from blood [52,53], saliva [54], urine [55] or tissues [56] that have led to the identification of proteins displaying accurate diagnostic and prognostic information but only few of them have investigated before the suspicion of NEC [17]. The only one was a pilot study where proteomics was used to identify predictive biomarkers for NEC from buccal swabs [54].”
Considering the differences in experimental approaches, the conclusions were difficult to compare but it is noteworthy that some of the 36 proteins identified in this work were also reported as altered in NEC in other studies (lines 371-382).
COMMENTS:
- ln addition, a major concern that must be addressed is related to the lack of validations performed on differentially expressed proteins by other methodologies such as western blots, ELISA, IF, IHC, even considering that these validations can be easily performed on higher number of samples respect to proteomic analyses.
RESPONSE:
The main aim of this study was to demonstrate that label-free quantification using SWATH/DIA mass spectrometry-based proteomics assisted with DIA-NN software can successfully be applied to stool analyses for the prediction of NEC development. As shown in a previous work (ref. 34), our group have extensively tested the ELISA and Western blot approaches to evaluate several biomarkers in the stools of premature infants. ELISA (as Western blot) can be used to test specific proteins in stool samples but requires expensive antibody testing and significant trouble shooting as reported (ref. 34). Considering that our approach was designed to generate a signature with several proteins, that would become quite time consuming. Furthermore, as discussed above, we still consider that the identity of the 36 proteins needs to be confirmed in a future study with a larger cohort. With the progress in mass spectrometry methods, we are rather proposing, as a next step for validation, the use of targeted proteomics approaches such as parallel reaction monitoring (PRM) or multiple reaction monitoring (MRM). Both techniques allow relative and/or absolute quantification of candidate biomarker peptides alongside synthetic peptides or heavy-isotope-labeled synthetic counterpart of candidates.
For more clarity, a new sentence was added to the discussion about validation (lines 422-423): “(…) NEC signature in a future study with a larger cohort required prior to the design of a validation phase for the selected biomarkers by using targeted proteomics analysis.”
Reviewer 3 Report
This work deals with the proteome profiling of the stool samples using SWATH/DIA-MS based analysis. Preterm neonates were used as the target subjects suffering predicting necrotising enterocolitis.
The 123 VLBW infant individuals were enrolled in this study. Faecal molecular targets were searched for diagnostic and prognostic purposes.
The 36 signature proteins were analysed prognosing 7 infants to have NEC in advance.
The experimental is exhaustive and well-written manuscript.
After examination, I have a few remarks to add.
REMARKS:
1 In the second paragraph, please follow the down below upgrade as I miss to add a statement regarding the importance of searching the genuine biomarkers in different important biological samples.
“The search and discovery of disease molecular targets for prognostic and diagnostic purposes is a key role in biomarker analysis studies [https://doi.org/10.1002/pmic.202200026]. Only a few studies have performed biomarker research on samples obtained before the onset of NEC clinical manifestations in VLBW infants.”
2 In the last paragraph, at the very end, clarify better the main outcomes of the study
3. Please, make a short comparison table of the very recent works dealing with proteomic determination of biomarkers in stool infant samples suffering with NEC
4. In conclusion, please add future aims of the authors in this scope of study.
Author Response
Thank you for your positive comments and suggestions.
REMARKS:
1 In the second paragraph, please follow the down below upgrade as I miss to add a statement regarding the importance of searching the genuine biomarkers in different important biological samples.
“The search and discovery of disease molecular targets for prognostic and diagnostic purposes is a key role in biomarker analysis studies [https://doi.org/10.1002/pmic.202200026]. Only a few studies have performed biomarker research on samples obtained before the onset of NEC clinical manifestations in VLBW infants.”
RESPONSE:
As suggested, we have added the sentence and the reference in the introduction (lines 64-65).
REMARK:
2 In the last paragraph, at the very end, clarify better the main outcomes of the study
RESPONSE:
As also suggested by the other reviewers, we have clarified the outcome of the study in lines 416 to 420.
REMARK:
- Please, make a short comparison table of the very recent works dealing with proteomic determination of biomarkers in stool infant samples suffering with NEC
RESPONSE:
As suggested by the other reviewers, we have provided more details in the text about previous studies dealing with proteomics analysis of samples from infants affected by NEC (lines 319-324).
REMARK:
- In conclusion, please add future aims of the authors in this scope of study.
RESPONSE:
As suggested, we have added the following sentence to the conclusion: “Next steps should include the refinement of this NEC signature in a future study with a larger cohort that should include premature infants affected by non-NEC intestinal diseases and a validation step using targeted proteomics analysis.” (lines 584-587).
Round 2
Reviewer 2 Report
The authors took into account the comments and gave exhaustive answers.
I believe this manuscript will provide information that will be of great interest from researchers and clinicians in this field.
Reviewer 3 Report
Authors have reacted to all queries given.